**Data Availability Statement:** All relevant data are within the manuscript and its Supporting Information files.

**Funding:** This work was supported by the National Research Foundation of Korea (NRF) grant funded

# Assessing the impact of apnea duration on the relationship between obstructive sleep apnea and hearing loss

Yong Seok Jo[1], Jeon Mi Lee[2]*

1 Department of Otorhinolaryngology-Head and Neck Surgery, Asan Medical Center, University of Ulsan College of Medicine, Seoul, Korea, 2 Department of Otorhinolaryngology, Ilsan Paik Hospital, Inje University College of Medicine, Goyang, Korea

* entmeowmiya@gmail.com

## Abstract

### Objectives

The relationship between obstructive sleep apnea (OSA) and hearing loss (HL) remains uncertain. This study aimed to investigate the relationship between OSA and HL, and to identify which factors play a key role.

### Methods

A retrospective review was conducted of 90 subjects diagnosed with OSA. These subjects underwent overnight polysomnography (PSG) and pure-tone audiometry at a single institution from February 2014 to November 2023. Hearing evaluations involved the comparison of OSA subjects with a non-OSA group, identified through national data utilizing the STOP-BANG questionnaire (SBQ) and age-sex 1:1 matching. Subsequently, individuals with OSA were categorized into HL and non-HL groups. Comparisons were made to ascertain differences in PSG parameters, followed by regression analysis to assess their actual impact.

### Results

The OSA group exhibited elevated hearing thresholds across all frequencies compared to the non-OSA group. Furthermore, classification of OSA subjects into the HL and non-HL groups revealed a statistically significant increase in apnea duration in the HL group for all-frequency and high-frequency cases (p = 0.038, 0.006). Multiple linear regression analysis, adjusting for age and sex, revealed a significant influence of apnea duration on HL in both all-frequency and high-frequency cases (ß = 0.404, p = 0.002; ß = 0.425, p = 0.001).

### Conclusion

These findings underscore the significant association between OSA and reduced auditory function, with apnea duration standing out as a crucial factor contributing to hearing loss. Our results suggest that prolonged apnea duration may be a marker of chronic hypoxic

by the Korea government (MSIT) (No. 2022R1F1A1071824) awarded to JML.

**Competing interests:** The authors have declared that no competing interests exist.

damage in patients with OSA, further clarifying its potential role in the development of hearing loss.

## Introduction

Obstructive sleep apnea (OSA) is defined as either complete or partial airway blockage during sleep, accompanied by decreased oxygen levels or awakening [1], and is highly prevalent globally, ranging from 9–38% [2]. Airway blockage induces hypoxia, and it may directly cause oxidative imbalance by producing reactive oxygen species and triggering an inflammatory cascade. This condition could develop endothelial dysfunction, resulting in developing cardiovascular and cerebrovascular diseases [3]. Furthermore, this phenomenon occurs more severely in longer respiratory events than in shorter events, most probably due to the higher hypoxic burden associated with longer respiratory events [4].

However, despite the growing prevalence of OSA [5], comprehensive data on whether hearing level changes occur in patients with OSA, and the patterns these changes follow, remain scarce. This gap highlights the need for further investigation to elucidate the auditory effects of OSA. Numerous studies have explored the potential association between OSA and hearing loss (HL). Researchers have focused on the impact of chronic intermittent hypoxic damage on the cochlea, which is particularly susceptible to hypoxia because of the absence of collateral circulation [6]. Nonetheless, consensus on the definitive association between OSA and HL, or the precise factors contributing to HL, remains elusive. Chauhan et al. suggested that OSA might contribute to HL, particularly in the high-frequency range exceeding 8 kHz [7]. Chopra et al. also reported a correlation between the severity of OSA and the degree of HL [8]. They emphasized that the lowest oxygen saturation is the key influencer of HL, consistent with Seo et al.'s report [9]. In a study involving 28 patients with OSA and 15 without OSA, Vorlová et al. found a correlation between OSA severity and high-frequency HL (HFHL) above 4 kHz [10]. Conversely, Lu et al. found no association between OSA and HL and identified a link to the risk of tinnitus [11]. İriz et al. and Hwang et al. agreed that no association exists between OSA and hearing level; however, they suggested that OSA exhibits a stronger association with central auditory processing rather than peripheral hearing level [12, 13]. Furthermore, a previous study indicated that snoring, rather than OSA, induces HFHL [14].

Despite numerous studies conducted to date, establishing a definitive relationship between OSA and HL remains challenging. Although overnight polysomnography (PSG) remains the gold standard for diagnosing OSA, its implementation is hindered by inconvenience and high cost, leading to a limited number of studies. Lisan et al. associated OSA with HL using the Berlin questionnaire for OSA diagnosis [15], whereas Li et al. and Lee et al. employed the STOP-BANG questionnaire (SBQ) [16, 17]. Chopra et al. utilized a portable PSG, potentially leading to limited variables and low accuracy [8]. Despite several studies using overnight PSG, the sample sizes were small. İriz et al. involved only 21 patients, and Seo et al. had 41 participants [9, 12]. Vorlová et al. studied 43 patients, with 15 identified as non-OSA [10]. Several studies used simplified hearing tests. Some studies conducted pure-tone audiometry and assessed hearing as a bivariate or stratified variable rather than a continuous variable [8, 12, 16]. Others did not measure hearing below 1 kHz [10] or had limited assessments over 4 kHz [11, 12, 16]. Consequently, studies concurrently conducting overnight PSG and reliable pure-tone audiometry have been limited to a small number of participants.

Therefore, in this study, we aimed to address the constraints observed in previous studies by focusing on several individuals diagnosed with OSA through overnight PSG and accurate pure-tone audiometry assessing from 0.25 to 8 kHz. Thus, by selecting participants with proper diagnosis and strict criteria, we believe that in this study, we could identify the true relationship between OSA and HL. We also aimed to determine which OSA characteristics affect hearing loss using PSG parameters.

## Materials and methods

### Study design and participants

This is a retrospective chart review study. This study included patients who underwent simultaneous overnight PSG and pure-tone audiometry within 1 year between February 2014 and November 2023. These are patients who voluntarily visited the hospital for diagnosis because they were suspected of having OSA. However, we excluded patients who were not diagnosed with OSA due to unmet criteria, such as an apnea-hypopnea index (AHI) < 5. Patients who underwent pure-tone audiometry due to pathological conditions like sudden hearing loss, otitis media, and trauma were also excluded. Cases in which pure-tone audiometry did not include high-frequency testing or those in which portable PSG was used were also excluded.

Furthermore, to perform a 1:1 age-sex matching with data from the Korean National Health and Nutrition Examination Survey (KNHANES), patients outside the age range of 40 to 80 years were not considered. Overall, 200 participants were initially reviewed, and based on the inclusion and exclusion criteria, 90 patients were enrolled and included in the OSA group. The normal control group was selected based on data from the KNHANES conducted between 2019 and 2020 by the Disease Control Headquarters to generate nationwide statistics on the health and nutritional status of Koreans. The survey included the SBQ, a validated screening tool for OSA, and pure-tone audiometry ranging from 0.5–8 kHz. Those who completed both the SBQ and pure-tone audiometry and were classified as low risk for OSA were defined as the non-OSA group and matched 1:1 by age and sex to the OSA group.

This study was approved by the institutional review board of Ilsan Paik Hospital, Inje University College of Medicine (approval number: 2022-12-017 and 2022-12-023). Because of the retrospective design of the study, informed consents were waived. The study was performed in accordance with the tenets of the Declaration of Helsinki and Good Clinical Practice guidelines. Data were accessed for research purposes between December 2022 and December 2023. Authors had no access to information that could identify individual participants during or after data collection.

### Overnight polysomnography

Trained technicians in an independent sleep laboratory conducted overnight PSG using a 14-channel PSG system (PSG-1100; NIHON KOHDEN, Japan). The PSG setup included an electroencephalogram (full 10–20 montage), electrooculogram, electrocardiogram, airflow monitoring, transcutaneous oxygen saturation, end-tidal carbon dioxide measurement, respiratory effort monitoring, snoring detection, chin-leg electromyogram, and a body position sensor. The examinations were performed by monitoring technicians in a control room.

A sleep-certified physician interpreted PSG findings in accordance with the American Academy of Sleep Medicine guidelines. Apnea was characterized as a signal decrease of more than 90% from baseline lasting over 10 s, whereas hypopnea was identified as a signal reduction exceeding 30% from baseline for over 10 s, accompanied by a desaturation of over 3% and an associated arousal. The AHI was calculated as the sum of apnea and hypopnea episodes per hour. For each respiratory event, the duration was measured, and the longest duration of

apnea was noted as the apnea maximum length, the longest duration of hypopnea as the hypopnea maximum length, and their sum as the total maximum length. The diagnostic criteria for OSA were defined as an AHI of ≥5, with severity categorized as mild (5≤AHI<15), moderate (15≤AHI<30), and severe (AHI≥30).

## Audiometric evaluation

A certified audiologist conducted pure-tone audiometry within a soundproof booth, assessing frequencies of 0.25, 0.5, 1, 2, 4, 6, and 8 kHz using a MADSEN Astera[2] (GN Otometrics, Denmark). The frequency order was randomized, and tests were administered using ascending and descending methods from 0 to 100 dB in 5 dB steps. The pure-tone average (PTA) was defined as the mean hearing threshold at 0.5, 1, 2, and 4 kHz, whereas the high-frequency pure-tone average (HF PTA) was defined as the mean hearing threshold at 2, 4, and 8 kHz. PTA and HF PTA from each ear were compared, and a better hearing level was selected for analysis to exclude patients with pathologically damaged hearing.

## Physical measurement

Body mass index (BMI) was calculated by dividing the body weight (kg) by the square of the height (m). Neck circumference (NC) was measured at the level of the 7th cervical spine, whereas abdomen circumference (AC) was assessed at the midpoint between the costal margin and the iliac crest. Hip circumference (HC) was measured around the widest area of the buttocks.

## Statistical analysis

All statistical analyses were performed using the SPSS version 23 for Windows (IBM Corp., Armonk, NY, United States). The Kolmogorov-Smirnov and Shapiro-Wilk tests were used to examine whether the variables satisfied normality. When the variables were compared between the two groups, the t-test and Mann-Whitney U test were used. When the variables were compared among the three groups, the Kruskal-Wallis test was conducted, followed by subsequent post-hoc tests using the Mann-Whitney test. P-values were adjusted by dividing them by the number of post-hoc test trials using Bonferroni's method. The chi-square and Fisher's exact tests were used to compare proportions between groups. Multiple linear regression was conducted to control for the variables using the stepwise selection method. The dependent variables were set as PTA and HF PTA.

Furthermore, to investigate the impact of PSG parameters on hearing and adjust for other factors, the independent variables included age, sex, and PSG parameters such as AHI, lowest O2 saturation, maximum apnea length, and maximum hypopnea length. Additionally, physical measurements such as BMI, NC, AC, and HC were included as independent variables. The stepwise selection method was selected for the analysis. Statistical significance was set at p < 0.05.

## Results

### Characteristics of the study population

A total of 90 participants were included in this study. The mean age was 61.1 ± 12.6 years (range, 22–86 years), and 72 were male. According to the criteria, patients were categorized as mild (n = 9), moderate (n = 30), or severe OSA (n = 51). The average ages were 54.7 ± 10.1, 64.5 ± 11.9, and 60.3 ± 13.0 years for the mild, moderate, and severe groups, respectively. Notably, the highest average age was observed in the moderate group; however, no statistically

**Table 1. Characteristics of the study population according to the severity of obstructive sleep apnea.**

| | | Total | Mild | Moderate | Severe | P-value[a] | G1vs.G2[b] | G2vs.G3[b] | G1vs.G3[b] |
|---|---|---|---|---|---|---|---|---|---|
| **Number of participants** | | 90 | 9 | 30 | 51 | | | | |
| **Age (years)** | | 61.1 ± 12.6 | 54.7 ± 10.1 | 64.5 ± 11.9 | 60.3 ± 13.0 | 0.098 | | | |
| **Sex** | **Male** | 72 (80%) | 8 (89%) | 21 (70%) | 43 (84%) | 0.233 | | | |
| | **Female** | 18 (20%) | 1 (11%) | 9 (30%) | 8 (16%) | | | | |
| **Physical measurements** | **BMI (kg/m²)** | 25.8 ± 4.0 | 23.5 ± 1.5 | 24.9 ± 3.5 | 26.8 ± 4.3 | **0.007** | 0.215 | 0.032 | **0.005** |
| | **NC (cm)** | 44.5 ± 5.4 | 41.7 ± 4.3 | 43.0 ± 5.3 | 45.9 ± 5.2 | **0.018** | 0.636 | **0.020** | 0.032 |
| | **AC (cm)** | 97.9 ± 7.5 | 93.9 ± 3.7 | 96.2 ± 7.4 | 100.0 ± 7.6 | **0.022** | 0.379 | 0.054 | **0.015** |
| | **HC (cm)** | 99.4 ± 7.3 | 95.1 ± 3.5 | 98.0 ± 7.5 | 100.9 ± 7.3 | **0.037** | 0.152 | 0.250 | **0.010** |
| **PSG parameters** | **AHI (/hr)** | 37.5 ± 21.9 | 8.7 ± 1.7 | 22.1 ± 3.9 | 51.6 ± 18.7 | **<0.001** | **<0.001** | **<0.001** | **<0.001** |
| | **RDI (/hr)** | 39.7 ± 21.9 | 11.8 ± 3.6 | 24.5 ± 4.3 | 53.5 ± 19.2 | **<0.001** | **<0.001** | **<0.001** | **<0.001** |
| | **Lowest O2 saturation (%)** | 81.7 ± 9.8 | 89.4 ± 2.4 | 85.6 ± 6.5 | 78.4 ± 10.6 | **<0.001** | 0.110 | **<0.001** | **<0.001** |
| | **Apnea maximum length (s)** | 35.5 ± 22.9 | 20.9 ± 18.0 | 29.9 ± 17.5 | 41.0 ± 25.0 | 0.051 | | | |
| | **Hypopnea maximum length (s)** | 85.6 ± 37.5 | 108.6 ± 45.2 | 88.0 ± 37.3 | 80.8 ± 36.0 | 0.137 | | | |
| | **Total maximum length (s)** | 121.0 ± 41.8 | 129.4 ± 41.4 | 117.9 ± 41.2 | 121.8 ± 42.8 | 0.699 | | | |
| **Pure-tone audiometry (dB HL)** | **250 Hz** | 18.5 ± 11.4 | 11.7 ± 6.1 | 22.7 ± 14.2 | 17.3 ± 9.4 | **0.038*** | **0.008** | 0.060 | 0.045 |
| | **500 Hz** | 19.2 ± 12.2 | 14.4 ± 6.8 | 24.0 ± 15.2 | 17.3 ± 10.1 | 0.079 | | | |
| | **1 kHz** | 23.1 ± 15.6 | 13.3 ± 7.5 | 28.8 ± 19.1 | 21.5 ± 13.2 | **0.040*** | **0.012** | 0.063 | 0.037 |
| | **2 kHz** | 28.3 ± 18.9 | 16.7 ± 9.4 | 33.2 ± 20.7 | 27.5 ± 18.2 | 0.075 | | | |
| | **3 kHz** | 34.9 ± 22.4 | 25.0 ± 14.4 | 38.3 ± 22.2 | 34.6 ± 23.5 | 0.281 | | | |
| | **4 kHz** | 39.1 ± 22.8 | 27.8 ± 18.4 | 42.8 ± 22.0 | 38.9 ± 23.6 | 0.183 | | | |
| | **6 kHz** | 49.9 ± 24.2 | 37.8 ± 17.9 | 54.2 ± 24.4 | 49.5 ± 24.7 | 0.174 | | | |
| | **8 kHz** | 56.7 ± 25.4 | 48.9 ± 13.4 | 60.8 ± 26.8 | 55.6 ± 26.2 | 0.386 | | | |
| | **PTA** | 27.4 ± 15.9 | 18.1 ± 7.1 | 32.2 ± 18.2 | 26.3 ± 14.8 | 0.107 | | | |
| | **HF PTA** | 41.4 ± 20.8 | 31.1 ± 11.9 | 45.6 ± 21.7 | 40.7 ± 21.0 | 0.223 | | | |

G1, mild group of OSA; G2, moderate group of OSA; G3, severe group of OSA; BMI, body mass index; NC, neck circumference; AC, abdomen circumference; HC, hip circumference; PSG, polysomnography; AHI, apnea hypopnea index; RDI, respiratory disturbance index; PTA, pure-tone average; HF PTA, high-frequency pure-tone average

[a]p-value < 0.05, by Kruskal-Wallis test

[b]p-value < 0.017 (0.05/3), by Mann-Whitney test; Bonferroni's method

significant differences were found among the groups. Similarly, sex ratios were comparable among the groups. BMI, NC, AC, and HC exhibited higher values with higher severity, and these differences were statistically significant (**Table 1**).

## Polysomnography (PSG) parameters

As the severity of OSA increased, the AHI and respiratory disturbance index (RDI) values increased, and the lowest oxygen saturation decreased with significance (p<0.001, both). Apnea maximum length increased with severity (20.9 ± 18.0, 29.9 ± 17.5, and 41.0 ± 25.0), approaching statistical significance (p = 0.051). However, no significant differences were found in hypopnea or total maximum length between the groups (**Table 1**).

## Hearing thresholds

Hearing thresholds showed elevated values across all frequencies and mean values, including PTA and HF PTA, in the order of the mild, severe, and moderate groups. Statistically

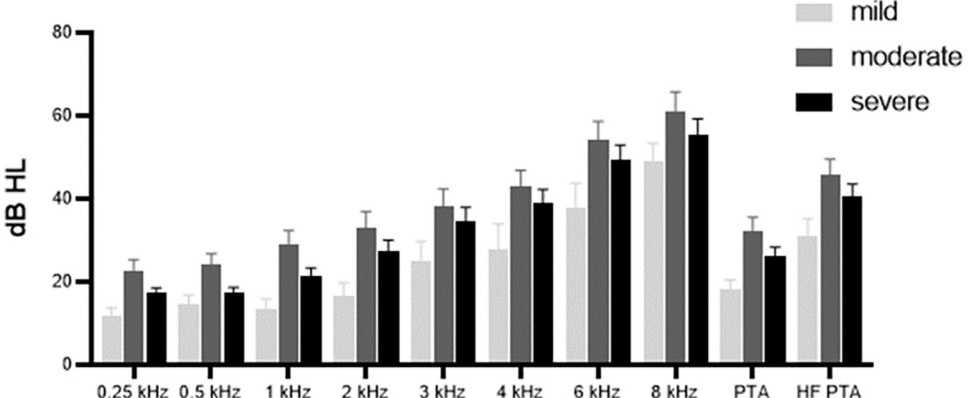

**Fig 1. Hearing thresholds across OSA severity.** Hearing thresholds showed elevated values across all frequencies and mean values in the order of the mild, severe, and moderate groups. Statistically significant differences were only noted between the mild and moderate groups at 0.25 and 1 kHz (p = 0.008 and 0.012).

significant differences were noted between the three groups only at 0.25 and 1 kHz (p = 0.038 and 0.040). Subsequent post-hoc tests demonstrated significant differences between the mild and moderate groups at 0.25 and 1 kHz (p = 0.008 and 0.012). However, no significant differences in hearing thresholds were found across all frequencies when comparing the moderate and severe groups (**Table 1** and **Fig 1**).

## Hearing level according to the presence of OSA

We compared hearing levels between the OSA and non-OSA groups and those categorized as low-risk for OSA using the SBQ from the KNHANES data.

The OSA group showed significantly worse hearing levels compared to the non-OSA group across all frequencies. Statistical significances were observed at frequencies of 0.5, 1, 2, and 8 kHz (p = 0.032, 0.002, 0.005, and 0.013, respectively), except for 4 kHz. Consequently, PTA and HF PTA also showed worse values in the OSA group, which were 27.4 ± 15.9 dB and 41.4 ± 20.8 dB, compared to the non-OSA group, which were 22.3 ± 12.3 dB (p = 0.016) and 34.9 ± 17.3 dB (p = 0.024) (**Table 2** and **Fig 2**).

## The factors contributing to hearing loss in patients with OSA

After establishing the impact of OSA on hearing, we attempted to identify specific factors of OSA contributing to HL. The OSA group was further classified into HL and non-HL groups

**Table 2. Comparison of hearing thresholds between OSA and non-OSA groups.**

|  | OSA (dB HL) | Non-OSA (dB HL) | p-value |
|---|---|---|---|
| **500 Hz** | 19.2 ± 12.2 | 15.6 ± 10.5 | **0.032** |
| **1 kHz** | 23.1 ± 15.6 | 16.2 ± 13.1 | **0.002** |
| **2 kHz** | 28.3 ± 18.9 | 21.2 ± 14.4 | **0.005** |
| **4 kHz** | 39.1 ± 22.8 | 36.0 ± 21.6 | 0.348 |
| **8 kHz** | 56.7 ± 25.4 | 47.4 ± 23.9 | **0.013** |
| **PTA** | 27.4 ± 15.9 | 22.3 ± 12.3 | **0.016** |
| **HF PTA** | 41.4 ± 20.8 | 34.9 ± 17.3 | **0.024** |

OSA, obstructive sleep apnea; PTA, pure-tone average; HF PTA, high-frequency pure-tone average

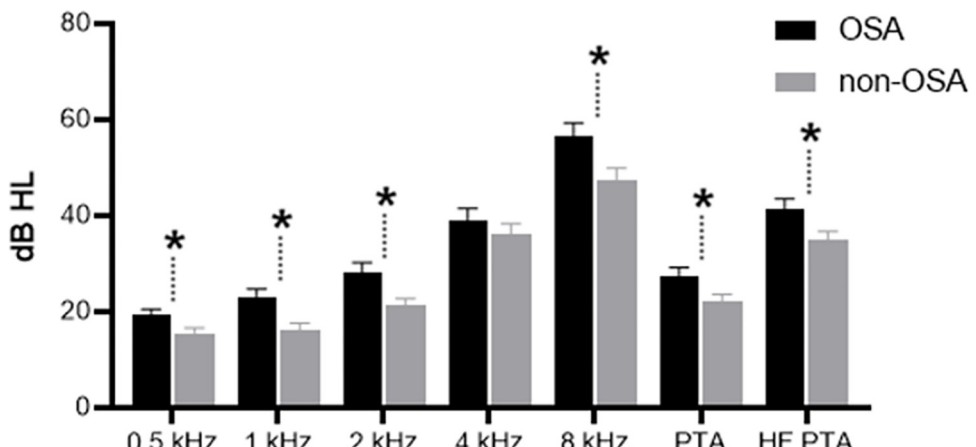

**Fig 2. Comparison of hearing thresholds between OSA and non-OSA groups.** The OSA group showed significantly worse hearing levels compared to the non-OSA group across all frequencies except at 4 kHz.

based on 40 dB, and PSG parameters were compared. PTA and HF PTA were used as standards (**Table 3**).

## Differences in PSG parameters between HL and non-HL groups

When the OSA group was categorized into two groups based on PTA values, 69 patients were classified into the HL group. Comparisons of PSG parameters revealed a significantly longer apnea maximum length in the HL group (44.8 s) compared to the non-HL group (32.7 s) (p = 0.038). Although tendencies towards longer hypopnea and total maximum length were observed in the HL group, these differences did not reach statistical significance. Additionally, despite higher AHI and RDI values in the HL group than in the non-HL group, these differences were not statistically significant (**Table 3**).

## Differences in PSG parameters between HFHL and non-HFHL groups

Similarly, after dividing the OSA group into HFHL and non-HFHL groups, 48 patients were classified into the HFHL group. The HFHL group also showed a significantly longer apnea maximum length than the non-HFHL group (p = 0.006). Additionally, the HFHL group showed higher values of hypopnea, total maximum length, AHI, and RDI than the non-HFHL group; however, these differences were not statistically significant (**Table 3**).

**Table 3. Comparison of PSG parameters between HL and non-HL groups and HFHL and non-HFHL groups.**

|  | HL | non-HL | p-value | HFHL | non-HFHL | p-value |
|---|---|---|---|---|---|---|
| N (%) | 69 (77%) | 21 (23%) |  | 48 (53%) | 42 (47%) |  |
| AHI (/hr) | 42.9 ± 24.8 | 35.8 ± 20.8 | 0.221 | 39.0 ± 23.9 | 36.1 ± 20.0 | 0.532 |
| RDI (/hr) | 46.7 ± 26.2 | 37.5 ± 20.2 | 0.123 | 42.0 ± 24.3 | 37.7 ± 19.7 | 0.362 |
| Lowest O2 saturation (%) | 82.2 ± 10.7 | 81.6 ± 9.6 | 0.301 | 81.6 ± 10.4 | 81.8 ± 9.4 | 0.912 |
| Apnea max. length (s) | 44.8 ± 25.9 | 32.7 ± 21.4 | **0.038** | 43.1 ± 23.9 | 29.5 ± 20.5 | **0.006** |
| Hypopnea max. length (s) | 87.1 ± 44.8 | 85.1 ± 35.4 | 0.841 | 86.8 ± 37.4 | 84.6 ± 38.0 | 0.793 |
| Total max. length (s) | 131.8 ± 48.2 | 117.8 ± 39.4 | 0.190 | 129.8 ± 43.6 | 114.1 ± 39.3 | 0.082 |

PSG, polysomnography; HL, hearing loss; HFHL, high-frequency hearing loss; AHI, apnea hypopnea index; RDI, respiratory disturbance index

**Table 4. Factors affecting hearing level in multiple linear regression analysis.**

| Independent variables | Dependent variables | PTA | | HF PTA | |
|---|---|---|---|---|---|
| | | ß | P-value | ß | P-value |
| Age | | 0.584 | **<0.001** | 0.602 | **<0.001** |
| Sex | | | .587 | | .326 |
| PSG parameter | AHI | | .859 | | .337 |
| | Lowest O2 saturation | 0.296 | **0.023** | 0.273 | **0.029** |
| | Apnea max. length | 0.404 | **0.002** | 0.425 | **0.001** |
| | Hypopnea max. length | | .098 | | .277 |
| Physical measurement | BMI | | .383 | | .395 |
| | NC | | .576 | | .550 |
| | AC | | .741 | | .778 |
| | HC | | .113 | | .175 |

PTA, pure-tone average; HF PTA, high-frequency pure-tone average; AHI, apnea hypopnea index; BMI, body mass index; NC, neck circumference; AC, abdomen circumference; HC, hip circumference

## Factors associated with hearing loss in patients with OSA

Although apnea maximum length showed a significant difference between the HL and non-HL groups, additional analyses were performed to determine the factors associated with HL in patients with OSA. We tried to investigate the specific impact of these factors while controlling for other influential factors. Besides age and sex, which are well-known factors affecting both OSA and hearing, every accessible variable, such as BMI, NC, AC, and HC, as well as PSG parameters including AHI, lowest oxygen saturation, apnea maximum length, and hypopnea maximum length, were considered. This analysis was performed for both the PTA and HF PTA samples.

A stepwise multiple linear regression demonstrated that age was the most significant influencing factor on HL (ß = 0.584 for PTA and ß = 0.602 for HF PTA). Apnea maximum length appeared as the second most influencing factor (ß = 0.404 for PTA, and ß = 0.425 for HF PTA), followed by lowest oxygen saturation (ß = 0.296 for PTA and ß = 0.273 for HF PTA). The other variables had no significant effect on HL (**Table 4**). Pearson correlation analysis

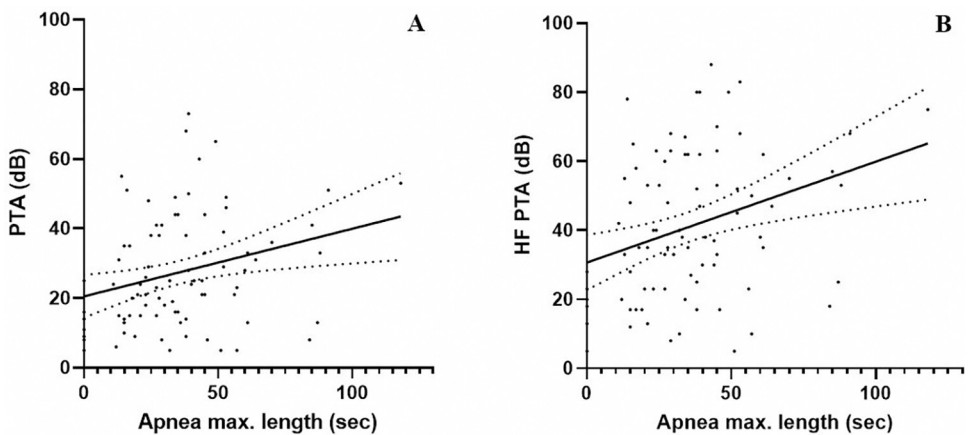

**Fig 3. The correlation between apnea maximum length and hearing level.** (A) Apnea maximum length showed a moderate correlation with PTA (r = 0.278). (B) Apnea maximum length showed a stronger correlation with HF-PTA (r = 0.317). PTA, pure-tone average; HF PTA, high-frequency pure-tone average.

showed that apnea maximum length had a correlation coefficient (r) of 0.278 and 0.317 with PTA and HF PTA, respectively (**Fig 3**).

## Discussion

In this study, we aimed to investigate the association between OSA and HL and determine factors influencing HL. Our findings suggest that OSA is associated with HL, highlighting the need to give closer attention to hearing levels in patients with OSA. The significant impact of apnea duration on hearing loss, even after controlling for other factors, indicates that prolonged apnea episodes may damage the auditory system. Clinically, this underscores the importance of diagnosis and treatment of OSA, particularly in the early stages, to potentially prevent further hearing decline. These findings suggest that incorporating a greater focus on hearing level in OSA management strategies could be crucial for enhancing the overall quality of care and preventing further complications associated with hearing loss in patients with OSA.

Previous studies investigating the link between OSA and HL have yielded inconsistent results. Thus, this study focused on selecting participants diagnosed with OSA through overnight PSG, coupled with accurate hearing assessments. Although we managed to obtain a large number of participants, recruiting an adequate number of normal control participants posed challenges, as asymptomatic individuals typically do not undergo PSG. Additionally, given the substantial impact of both age and sex on OSA and HL [18–20], controlling for these factors is crucial. Hence, this study selected normal control participants by conducting 1:1 age- and sex-matching from a nationwide population based on the SBQ. The SBQ is a validated alternative to PSG for assessing OSA risk. While the reported values vary, the SBQ consistently exhibits high sensitivity for OSA diagnosis, ranging from 83.6–100.0% [21–24]. The high sensitivity and low false-negative value for the SBQ suggest that individuals classified as low-risk by the SBQ likely have a low probability of latent OSA.

Furthermore, this study included all frequency and high-frequency assessments due to the varying results of previous studies. Notably, some studies have claimed that OSA causes HL at all frequencies [8, 25], whereas others have reported HL only at high frequencies [7, 26]. Kayabasi et al. proposed that this difference was due to the severity of OSA [25]. They compared the hearing levels in 120 patients with OSA and found that although patients with moderate OSA showed hearing loss only at high frequencies, those with severe OSA showed hearing loss at all frequencies. Conversely, other studies claimed that the effect of snoring on hearing loss was observed at high frequencies rather than hypoxic damage due to OSA [14]. Ekin et al. compared the hearing levels between patients with OSA, the simple snoring group, and normal controls [27]. They found no difference in hearing levels between patients with OSA and the simple snoring group; however, the simple snoring group exhibited HL in extended high frequencies > 10 kHz compared with the other two groups. Based on the results, they argued that snoring, which was a continuous noise exposure, was the main cause of hearing loss, especially at high frequencies. However, the participants in this study were mostly those with moderate OSA; therefore, there may not have been much difference in their hearing levels compared with the normal control group. Furthermore, we did not consider OSA severity or snoring in the present study since we have focused more on strictly selecting participants and controlling other factors that may affect hearing levels. Therefore, future studies that consider OSA severity and snoring should further explore the relationship between OSA and hearing level.

In this study, the OSA group exhibited significantly worse hearing across all frequency ranges; however, no statistical significance was observed at 4 kHz. This is likely because the

present study was conducted in Korea, where several years of military service are mandatory. Noise-induced hearing loss associated with military service is frequently reported in Korean men, leading to substantial HL, particularly at approximately 4 kHz [28]. Therefore, considering the high proportion of men in this study, this additional noise effect might have contributed to the reduced statistical differences at this frequency. Consequently, our result confirmed that hearing levels were significantly worse across all frequencies in the OSA group compared with the normal control.

We found apnea duration, along with age and the lowest oxygen saturation level, as significant factors influencing HL among individuals with OSA. Several studies have attempted to determine the specific aspects of OSA with HL. Although many have concentrated on OSA severity, typically measured by the AHI, findings have been inconsistent. Some studies found a dose-dependent relationship between AHI and HL [8, 29], whereas others indicated no such relationship [11]. Moreover, alternative factors, such as the oxygen desaturation index (ODI) [30], lowest oxygen saturation level [9], and sleep time with oxygen saturation below 90% (ST90) [10], have been proposed as potential contributors. However, these parameters cannot accurately capture the prolonged hypoxic state responsible for HL. OSA, being a chronic condition, causes tissue damage through sustained hypoxia. Therefore, a parameter reflecting the duration of this hypoxic state is needed. PSG is a useful tool for examining the state of an individual during sleep and only reflects the moment of examination. PSG implements variable parameters, such as AHI, RDI, lowest oxygen saturation level, ST90, and ODI, to assess the hypoxic state. However, AHI, RDI, and ODI represent the frequency of hypoxic events and do not accurately reflect the duration of the hypoxic state. Additionally, individual variations in oxygen saturation baselines and instability, often influenced by underlying conditions like cardiopulmonary diseases, make it challenging to precisely evaluate the degree of hypoxia using specific saturation values. For instance, a study of patients with chronic obstructive pulmonary disease revealed that the minimum arterial oxygen saturation level during awake hours was consistently below 90%, even during rest or simple daily activities, such as walking or eating [31]. Furthermore, individuals with congestive heart failure experience fluctuations in oxygen saturation even during the waking periods [32]. Given the significant variability in oxygen saturation fluctuations, especially in patients with OSA and multiple comorbidities, accurately assessing chronic hypoxia based on specific saturation values, such as the lowest oxygen saturation or ST90, may pose challenges.

This study prioritized assessing the duration of hypoxic conditions during apnea episodes (apnea maximum length) rather than solely relying on the frequency of apnea events (AHI). This approach was adopted because determining the duration of OSA episodes is challenging. The duration of hypoxia has a major effect on nerve injury, more so than the frequency of occurrences. An animal study investigating cerebral damage due to ischemia found that neuronal death occurred after 5 min of bilateral common carotid blood flow obstruction and not under 2 min, suggesting that short-term hypoxia may protect against subsequent long-term hypoxia exposure [33]. Moreover, peripheral nerves are more resistant to ischemia than the brain tissue [34–36]. These findings help explain why patients with longer apnea maximum lengths exhibited more significant hearing loss in this study, suggesting that the duration of hypoxia primarily influences hearing loss in individuals with OSA.

This study had several limitations. Although efforts were made to select normal control participants by conducting a 1:1 age- and sex-matching from the low-risk OSA group based on a validated questionnaire, patients without OSA were not confirmed by overnight PSG. Although the SBQ exhibited high sensitivity, resulting in minimal undetected cases of OSA, this could still have posed a critical limitation. Another limitation was the lack of consideration for the impact of snoring. The PSG sensor employed in this study could not objectively

measure the snoring sound volume. Despite conflicting views on the relationship between snoring and HL, the potential cumulative impact of loud snoring, which can reach up to 100 dB [37, 38], might not have been adequately addressed in this study. Additionally, we did not consider underlying comorbidities or social histories such as alcohol consumption, smoking, and occupational histories. Notably, given reports of various chronic conditions, including cardiovascular disease, hypertension, and diabetes mellitus being associated with OSA and HL [16, 39–43], the failure to consider these factors could also be viewed as a limitation. Moreover, this limitation might extend to the absence of an assessment of the association between OSA and central HL, which includes measures such as speech audiometry, auditory temporal processing, and sequencing, as observed in other studies. Finally, because we did not differentiate between the apnea and hypopnea subtypes, a possibility exists that central or mixed sleep apnea was included. Nevertheless, given that the majority of sleep apnea cases are OSA [44], the expected impact is not expected to be substantial.

However, despite its limitations, this study provides valuable insights into the relationship between OSA and hearing loss, utilizing overnight PSG and precise pure-tone audiometry ranging from 250 to 8 kHz. The results clearly show that individuals with OSA have poorer hearing across all frequencies compared with those without OSA, and a significant correlation was found between apnea duration and hearing loss. Notably, this study highlights the potential of using apnea duration as an indicator of hypoxia-related auditory damage. These findings underscore the importance of early detection and intervention in patients with OSA to prevent further hearing loss, marking a crucial step toward understanding the complex link between sleep apnea and hearing impairment. However, future studies should address and overcome these limitations while exploring parameters capable of effectively illustrating the relationship between OSA and HL. Specifically, potential confounding variables, such as snoring sound, lifestyle factors, or additional comorbidities that may influence OSA severity and hearing, should be considered.

Moreover, recent attention has been drawn to the concept of 'Hypoxic burden', which refers to the cumulative impact of oxygen deprivation during sleep. Therefore, investigating the relationship between hypoxic burden and hearing loss could provide valuable insights into how intermittent hypoxia may contribute to auditory damage in patients with OSA. Thus, further exploration of this parameter may offer a more comprehensive understanding of the mechanisms driving the association between OSA and HL, as well as potential therapeutic targets.

## Supporting information

**S1 Data. The raw data of this study.**
(XLSX)

## Author Contributions

**Conceptualization:** Jeon Mi Lee.

**Data curation:** Jeon Mi Lee.

**Formal analysis:** Yong Seok Jo.

**Methodology:** Jeon Mi Lee.

**Supervision:** Jeon Mi Lee.

**Validation:** Yong Seok Jo.

**Visualization:** Yong Seok Jo.

**Writing – original draft:** Yong Seok Jo.

**Writing – review & editing:** Yong Seok Jo, Jeon Mi Lee.

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
