## [Decision Letter · Decision Letter 0]

1 Oct 2024

PONE-D-24-38438Assessing the impact of apnea duration on the relationship between obstructive sleep apnea and hearing lossPLOS ONE

Dear Dr. Lee,

Thank you for submitting your manuscript to PLOS ONE. After careful consideration, we feel that it has merit but does not fully meet PLOS ONE’s publication criteria as it currently stands. Therefore, we invite you to submit a revised version of the manuscript that addresses the points raised during the review process.

We look forward to receiving your revised manuscript.

Kind regards,

Giovanni Cammaroto

Academic Editor

PLOS ONE

Journal Requirements:

"This work was supported by the National Research Foundation of Korea (NRF) grant funded by the Korea government (MSIT) (No. 2022R1F1A1071824) to JML"

Reviewers' comments:

Reviewer's Responses to Questions

**Comments to the Author**

1. Is the manuscript technically sound, and do the data support the conclusions?

Reviewer #1: Yes

Reviewer #2: Yes

2. Has the statistical analysis been performed appropriately and rigorously? 

Reviewer #1: Yes

Reviewer #2: Yes

3. Have the authors made all data underlying the findings in their manuscript fully available?

Reviewer #1: Yes

Reviewer #2: Yes

4. Is the manuscript presented in an intelligible fashion and written in standard English?

Reviewer #1: Yes

Reviewer #2: Yes

5. Review Comments to the Author

Reviewer #1: Introduction:

- Give additional background on the clinical significance of investigating the connection between OSA and hearing loss.

- Clearly state why the length of apnea is a significant aspect. cite doi:10.3390/ijerph19042459

- At the conclusion of the introduction, provide a more detailed statement of the study's precise hypotheses or aims.

Methods:

- Give more information about the inclusion and exclusion criteria as well as patient recruitment.

- Justify the selection of 90 OSA patients as the sample size.

- Give a more thorough explanation of the statistical analysis, focusing on the multiple regression method.

Results:

- Use subheadings to make the results easier to understand and present them in a more structured manner.

- When applicable, provide confidence intervals in addition to p-values.

- Take into account include a graph illustrating the connection between the length of apnea and hearing loss.

Discussion:

- Talk about the findings' clinical implications in greater detail.

- Make a thorough comparison and contrast between the findings and earlier research on OSA and hearing loss. cite doi:10.12659/msm.897347.

- Go into further detail about the study's limitations, including any potential confounding variables that were overlooked.

- Make more precise suggestions for future lines of inquiry.

Conclusion

- Rewrite the conclusion to make it more impactful and succinct.

- Make a clearer emphasis on the study's significance and main innovative findings.

Reviewer #2: Very good work on controlling for factors with adequate matching.

Including a low-risk SBQ result in the control group as a criterion for acceptance shows the robustness of the methodology.

The conclusion of the abstract is misleading, which findings underscore the HL? The reserach findings underscore? or is the previously suspected HL on the OSA patients what is underscored ?

I recommend rewording this concept.

6. PLOS authors have the option to publish the peer review history of their article (what does this mean?). If published, this will include your full peer review and any attached files.

Reviewer #1: **Yes: **Antonino

Reviewer #2: No

---

## [Author Response · Author response to Decision Letter 0]

15 Nov 2024

We sincerely thank the reviewers for their detailed and thoughtful reviews. We have addressed your comments to the best of our ability and uploaded our responses in the "Response to Reviewers" file. We hope that this revision meets the standards for publication in the esteemed journal, PLOS ONE.

Thank you.

---

## [Editor Report · Decision Letter 1]

28 Nov 2024

Assessing the impact of apnea duration on the relationship between obstructive sleep apnea and hearing loss

PONE-D-24-38438R1

Dear Dr. Jeon Mi Lee,

We’re pleased to inform you that your manuscript has been judged scientifically suitable for publication and will be formally accepted for publication once it meets all outstanding technical requirements.

Kind regards,

Giovanni Cammaroto

Academic Editor

PLOS ONE
---

## [Editor Report · Acceptance letter]

3 Dec 2024

PONE-D-24-38438R1 

PLOS ONE

Dear Dr. Lee, 

I'm pleased to inform you that your manuscript has been deemed suitable for publication in PLOS ONE. Congratulations! Your manuscript is now being handed over to our production team.

Kind regards, 

on behalf of

Dr. Giovanni Cammaroto 

Academic Editor

PLOS ONE